# Association between chiropractic spinal manipulation for sciatica and opioid-related adverse events: A retrospective cohort study

Robert J. Trager [1,2,3]*, Zachary A. Cupler [4,5], Roshini Srinivasan[1,6], Elleson G. Harper[7], Jaime A. Perez[7]

1 Connor Whole Health, University Hospitals Cleveland Medical Center, Cleveland, Ohio, United States of America, 2 Department of Family Medicine and Community Health, School of Medicine, Case Western Reserve University, Cleveland, Ohio, United States of America, 3 Department of Biostatistics and Bioinformatics Clinical Research Training Program, Duke University School of Medicine, Durham, North Carolina, United States of America, 4 Physical Medicine & Rehabilitative Services, Butler VA Health Care System, Butler, Pennsylvania, United States of America, 5 Institute for Clinical Research Education, University of Pittsburgh School of Medicine, Pittsburgh, Pennsylvania, United States of America, 6 School of Medicine, Duke University, Durham, North Carolina, United States of America, 7 Clinical Research Center, University Hospitals Cleveland Medical Center, Cleveland, Ohio, United States of America

* Robert.Trager@Duke.edu

**Data Availability Statement:** The minimal, de-identified, aggregated data for baseline characteristics, our primary outcome, and plots of propensity score density and cumulative incidence

## Abstract

### Background

Patients receiving chiropractic spinal manipulation (CSM) for spinal pain are less likely to be prescribed opioids, and some evidence suggests that these patients have a lower risk of any type of adverse drug event. We hypothesize that adults receiving CSM for sciatica will have a reduced risk of opioid-related adverse drug events (ORADEs) over a one-year follow-up compared to matched controls not receiving CSM.

### Methods

We searched a United States (US) claims-based data resource (Diamond Network, TriNetX, Inc.) of more than 216 million patients, yielding data ranging from 2009 to 2024. We included patients aged ≥18 years with sciatica, excluding those post-spine surgery, prior anesthesia, serious pathology, high risk of ORADEs, and an ORADE ≤ 1-year prior. Patients were divided into two cohorts: (1) CSM and (2) usual medical care. We used propensity score matching to control for confounding variables associated with ORADEs. Comparative outcomes were analyzed by calculating risk ratios (RRs) and 95% confidence intervals (CIs) for the incidence of ORADEs and oral opioid prescription between cohorts.

### Results

372,471 patients per cohort remained after matching. The incidence of ORADEs over 1-year follow-up was less in the CSM cohort compared to the usual medical care cohort (CSM: 0.09%; usual medical care: 0.30%), yielding an RR of 0.29 (95% CI: 0.25–0.32; _P_ <

are available in figshare (https://doi.org/10.6084/m9.figshare.25655964).

**Funding:** Employees of Connor Whole Health received support from the Elisabeth Severance Prentiss Foundation (Cleveland, OH) through general funding. This project is supported by the Clinical and Translational Science Collaborative of Northern Ohio, which is funded by the National Institutes of Health, National Center for Advancing Translational Sciences, Clinical and Translational Science Award grant, UM1TR004528. The content is solely the responsibility of the authors and does not necessarily represent the official views of the National Institutes of Health.

**Competing interests:** I have read the journal's policy and the authors of this manuscript have the following competing interests: Robert J. Trager acknowledges that he has received royalties as the author of two texts on the topic of sciatica. The other authors have declared no competing interests. This does not alter our adherence to PLOS ONE policies on sharing data and materials.

.00001). CSM patients had a lower risk of receiving an oral opioid prescription (RR of 0.68 [95% CI: 0.68–0.69; P < .00001]).

## Conclusions

This study found that adults with sciatica who initially received CSM had a lower risk of an ORADE compared to matched controls not initially receiving CSM, likely explained by a lower probability of opioid prescription. These findings corroborate existing practice guidelines which recommend adding CSM to the management of sciatica when appropriately indicated.

## Introduction

Opioids are narcotic analgesic medications that are often used to treat painful conditions such as sciatica, a radiating pain from the lumbar spine into the lower extremity most often caused by irritation of lumbosacral nerve root(s). Despite limited evidence of efficacy in this condition, opioids are frequently prescribed to treat sciatica [1–4]. Opioids may cause a range of adverse effects commonly including constipation, dizziness, and sedation, and less often, nausea and vomiting. In addition, opioids have the potential for misuse, long-term use, dependency, addiction, and respiratory depression leading to death [5]. Opioid related adverse drug events (ORADEs) are typically defined as moderate to severe adverse effects, such as opioid-related poisoning, overdose, and death [5, 6].

The United States' (US) Centers for Disease Control (CDC) and several recent clinical practice guidelines have discouraged prescribing opioids for acute musculoskeletal etiologies of low back pain [7, 8] while national health care systems have implemented opioid stewardship approaches involving interdisciplinary care and opioid safety initiatives [9]. As a result, the yearly percentage of US adults who received an opioid prescription has declined from 28% to 19% between 2008 and 2018 [10]. Despite this, ORADEs remain relatively common. In one cohort study including 5684 subjects, emergency department encounters or hospitalizations for ORADEs affected 1.7% of patients newly prescribed a long-acting opioid, yielding an incidence rate of two to six per 100 person-years [11]. Concerningly, the yearly incidence of opioid overdose deaths in the US has gradually increased over the past two decades, most recently 25 per 100,000 people in 2021, although this estimate also includes deaths from non-prescription opioid use, particularly illicit fentanyl [12], which influences mortality statistics. As a result, this value does not represent deaths strictly from prescribed opioids.

Chiropractic spinal manipulation (CSM) is a form of manual therapy directed to the joints of the spine. Given its clinical effectiveness for treating sciatica [13], it is recommended by clinical practice guidelines for treatment of this condition [14–18]. Prior studies suggest that receiving CSM for spinal pain is associated with lower rates of opioid prescription compared to usual medical care [19, 20] and potentially reduces risk of any adverse drug event compared to conventional medical management in adults [21, 22]. For example, one study found the risk of any outpatient adverse drug event was 51% lower over 12 months in those receiving CSM versus those who did not [21].

To date, no studies have specifically evaluated whether CSM is associated with a reduced likelihood of moderate to severe opioid-related adverse drug events (ORADEs) compared to usual medical care in any population. Our achieved aim was to examine this question using a real-world population of adults with sciatica. We hypothesized that adults receiving CSM for

sciatica would have a lower risk of ORADEs over 12 months' follow-up compared to propensity-matched controls not receiving CSM.

## Materials and methods

### Study design

The present retrospective cohort study incorporated active comparator features to minimize bias [23]. The findings are reported according to the Strengthening the Reporting of Observational Studies in Epidemiology (STROBE) [24].

Patients meeting the selection criteria from March 11, 2009, through March 11, 2023, were included, ensuring a one-year follow-up period for outcome ascertainment prior to the query date of March 11, 2024. The University Hospitals Institutional Review Board (IRB; Cleveland, OH, US) considers the current study methods of using fully anonymized, de-identified data from TriNetX (Cambridge, MA, US) acquired via the hospital's Clinical Research Center Honest Broker to meet criteria for 'Not Human Subjects Research' and did not require IRB review or patient consent.

The present study used data from a predominantly claims-based resource which includes real-world de-identified data from over 213 million patients (Diamond Network of TriNetX, Inc.). This network integrates open medical claims data from clearinghouses, representing primary care, outpatient, inpatient, specialty, and ancillary care settings, as well as pharmacy claims from switches. The data cover 99% of US health plans, including Commercial, Medicare, Medicaid, Veterans Affairs, and other payer types. The Diamond Network also links electronic health records data, and 44% of patients have these data available as of 2024. The network includes, but is not limited to, data relating to demographics, diagnoses, lab results, medications, procedures, and vital signs. These data are searched using standardized nomenclature including the International Classification of Diseases, 10th Edition codes (ICD-10) and Current Procedural Terminology (CPT) codes [25]. When searching older records, the TriNetX software automatically converts ICD-10 to ICD-9 codes. TriNetX adheres to the Health Insurance Portability and Accountability Act (HIPAA), only contains de-identified data, and anonymizes the health care organizations contributing data.

Our study followed our registered protocol [26], with the exception of using the TriNetX Diamond network which incorporates medical claims and includes a larger patient population. This change necessitated us to: (1) require patients to have a medical evaluation before and after the date of inclusion in the study to ensure completeness, and (2) use different codes used to identify gabapentin and skeletal muscle relaxants.

### Participants

**Eligibility criteria.** We included adults at least age 18 years, at the first occurrence of any diagnosis code of sciatica or lumbosacral radiculopathy (i.e., index date; S1 Table). This strategy aimed to minimize the variability in clinical presentation. We used the first recorded diagnosis code of sciatica as the index date rather than relying on a specific washout period, with the aim of minimizing imbalance between cohorts' durations of sciatica. To improve data completeness, we required patients to have a prior healthcare visit between one week and two years prior to the index date (CPT 1013625). To minimize loss to follow-up, we required patients to have at least one healthcare visit or have a recorded status of 'deceased' during the 1-year follow-up.

We excluded patients with serious pathology such as cauda equina syndrome, spinal infection, bleed, fracture, cancer, and alternate conditions causing spinal pain (e.g., spinal deformity, myelopathy) (S2 Table). We excluded individuals who were pregnant and therefore

unlikely to receive an opioid prescription [7], and those receiving palliative care who would be unlikely to receive CSM, and perhaps more likely to receive opioid therapy. We also excluded individuals with a much greater risk of ORADEs: those having an ORADE in the year preceding inclusion [7], opioid, cocaine, or stimulant use disorder, positive urine test for fentanyl, methamphetamine, or cocaine, or prescription of fentanyl, sufentanil, or hydromorphone (i.e., highly potent opioids) [27, 28], those taking medication assisted treatment for opioid use disorder (i.e., methadone and buprenorphine) [29], and those with any recent anesthesia or spine surgery [30]. Those with a 'do not resuscitate' status were excluded considering ORADEs could go undetected [31]. We excluded patients from the usual medical care cohort who received CSM on the index date of cohort eligibility.

## Variables

**Cohorts.** Patients were divided into two cohorts dependent on the treatment received on the index date of sciatica diagnosis: (1) CSM; those receiving any CPT code for this procedure (98940, 98941, 98942); and (2) usual medical care; those having an outpatient office visit (CPT: 1013625) and not receiving CSM on that date.

**Confounding variables.** We propensity matched patients to reduce bias [23], balancing confounders present within a year preceding and including the date of inclusion associated with risk of ORADEs, including previous prescription medications and opioids (S3 Table).

**Outcome.** We queried for a composite outcome of ORADEs, including both diagnosis codes and procedure codes specifying administration of naloxone for opioid overdose (S4 Table) [32, 33]. Diagnoses used in our outcome indicate moderate and severe ORADEs and fatal opioid-related overdoses [34], rather than mild adverse events [6, 35, 36]. We used a one-year follow-up window to account for variability in timing of ORADEs.

We did not use symptom-based codes such as dyspnea, nausea, vomiting, and constipation which may be unrelated to opioid use [36], and likewise avoided ICD-10 codes describing other or unspecified drug-related adverse events. We also did not examine heroin-related events considering this would predominantly reflect illicit drug use, or markers of long-term opioid use, misuse, or addiction, which may develop over the span of years and would require a larger sample of opioid-naïve patients.

As a sensitivity analysis, we plotted cumulative incidence of ORADEs per cohort to clarify timing of these events. We also calculated the RR of oral opioid prescription to determine whether an increase or decrease in RR of ADEs could be attributed to differential prescribing behavior.

## Statistical methods

We used built-in features of the online TriNetX analytics suite to compare baseline characteristics, using standardized mean difference (SMD>0.1) as a threshold for between-cohort imbalance. Calculation of propensity scores was conducted using logistic regression via Python (scikit-learn version 1.3 [Python Software Foundation, Delaware, US]). This model calculated the log odds of belonging to the usual medical care cohort, as a linear combination of matched covariates. The fitted model provided a propensity score for all patients which ranged from 0 to 1, representing the lowest to highest likelihood of receiving usual medical care. When balancing the cohorts, we conducted 1:1 greedy nearest neighbor matching, using a caliper width of 0.1 standard deviations pooled from the logit values of the propensity score.

Risk ratios (RRs) for ORADEs among patients with sciatica were derived by dividing the incidence proportion of ORADEs in the CSM cohort by the incidence proportion of ORADEs in the usual medical care cohort. To visualize total incidences, cumulative incidences with

locally weighted scatterplot smoothing, and propensity score densities, we used the ggplot2 package [37] in R (version 4.2.2, Vienna, AT [38]).

To further assess matching success, we calculated a post-matching RR for radiology procedures (CPT: 1010251) as a negative control outcome during the follow-up period [39], aiming for a between-cohort balance reflected by an RR of 0.73 to 1.38 [40].

### Required study size

We estimated a required total sample size of 10,032, aided by data from a previous study [21]. We used G*Power (Kiel University, DE), Z-tests for determining a difference between two independent proportions (0.01 vs. 0.004), alpha error of 0.05, power of 0.95, and an allocation ratio of one.

## Results

### Participants

Our query identified 372,471 patients in the CSM cohort and 2,090,255 in the usual medical care cohort before matching. After matching, there were 372,471 patients in each cohort. Before matching, patients in the CSM cohort were less often Hispanic or Latino, Black or African American, had a lower incidence of several comorbidities, including substance use disorders, and a lower incidence of prescription of several medications, including opioid analgesics, skeletal muscle relaxants, and gabapentin (SMD >0.1; Table 1). Following matching, key variables were optimally matched (SMD <0.1; Table 1). The incidence of naltrexone, which was not matched, was also similar between cohorts following matching of the other covariates (SMD <0.1; Table 1).

### Descriptive data

The mean number of facts (i.e., data points such as diagnoses and laboratory results) per patient per cohort was adequate (CSM: 719; usual medical care: 1,042). After propensity score matching, there was no meaningful difference between cohorts with respect to the proportion of patients with "unknown" demographic variables: unknown age (0% for both cohorts, SMD = 0), unknown sex (<1% for both cohorts, SMD = 0.004), and unknown ethnicity (75% in both cohorts, SMD = 0.006). After matching, the densities of both cohorts' propensity scores were closely superimposed, further highlighting the success of balancing matched confounding variables (Fig 1).

### Key results

The incidence of ORADEs over one year following the index date in adults with sciatica was lower in the CSM cohort compared to the usual medical care cohort (Table 2, Fig 2). After propensity matching, 0.09% (95% CI: 0.08–0.10%) of the CSM cohort had an ORADE, compared to 0.30% (95% CI: 0.29–0.32) of the usual medical care cohort, translating to an RR of 0.29 (95% CI: 0.25–0.32; P < .00001).

### Secondary outcomes

**Sensitivity analysis.** The cumulative incidence of ORADEs per cohort diverged immediately following the index date without overlap in the incidence curves or their 95% confidence intervals, showing a separation in incidence throughout the 1-year follow-up window (Fig 3).

The incidence of oral opioid prescription over one year following the index date for adults with sciatica was lower in the CSM cohort compared to the usual medical care cohort. After

**Table 1. Baseline characteristics before and after propensity matching.**

| Variable | Before matching | | | After matching | | |
|---|---|---|---|---|---|---|
| | CSM | Usual medical care | SMD | CSM | Usual medical care | SMD |
| N | 372471 | 2090255 | NA | 372471 | 372471 | NA |
| Demographics | | | | | | |
| Age, mean (SD) | 51.5 (15.8) | 52.5 (14.9) | 0.066 | 51.5 (15.8) | 51.4 (15.7) | 0.005 |
| Female, n (%) | 219762 (59%) | 1324739 (63%) | 0.090 | 219762 (59%) | 220391 (59%) | 0.003 |
| Male, n (%) | 152595 (41%) | 764830 (37%) | 0.090 | 152595 (41%) | 151991 (41%) | 0.003 |
| Hispanic or Latino, n (%) | 4234 (1%) | 63957 (3%) | 0.134 | 4234 (1%) | 4252 (1%) | <0.0001 |
| Not Hispanic or Latino, n (%) | 90376 (24%) | 591987 (28%) | 0.092 | 90376 (24%) | 89332 (24%) | 0.007 |
| Asian, n (%) | 540 (<1%) | 5541 (<1%) | 0.027 | 540 (<1%) | 477 (<1%) | 0.005 |
| Black or African American, n (%) | 4529 (1%) | 108600 (5%) | 0.227 | 4529 (1%) | 4645 (1%) | 0.003 |
| White, n (%) | 85263 (23%) | 491039 (23%) | 0.014 | 85263 (23%) | 83955 (23%) | 0.008 |
| Comorbidities, n (%) | | | | | | |
| Adverse socioeconomic and psychosocial circumstances | 3806 (1%) | 53255 (3%) | 0.115 | 3806 (1%) | 3247 (1%) | 0.015 |
| Alcohol related disorders | 3211 (1%) | 60113 (3%) | 0.149 | 3211 (1%) | 2968 (1%) | 0.007 |
| Chronic kidney disease | 7249 (2%) | 95258 (5%) | 0.148 | 7249 (2%) | 7007 (2%) | 0.005 |
| Chronic obstructive pulmonary disease | 8006 (2%) | 158164 (8%) | 0.254 | 8006 (2%) | 7962 (2%) | 0.001 |
| Diabetes mellitus | 37912 (10%) | 392601 (19%) | 0.246 | 37912 (10%) | 37536 (10%) | 0.003 |
| Diseases of liver | 6679 (2%) | 83893 (4%) | 0.133 | 6679 (2%) | 6269 (2%) | 0.008 |
| Hypertensive diseases | 87373 (23%) | 804219 (38%) | 0.329 | 87373 (23%) | 87509 (23%) | 0.001 |
| Mood disorders | 31189 (8%) | 366957 (18%) | 0.276 | 31189 (8%) | 31112 (8%) | 0.001 |
| Nicotine dependence | 13392 (4%) | 295931 (14%) | 0.378 | 13392 (4%) | 13669 (4%) | 0.004 |
| Osteoarthritis | 15029 (4%) | 178523 (9%) | 0.186 | 15029 (4%) | 14577 (4%) | 0.006 |
| Sedative, hypnotic, or anxiolytic related disorders | 201 (<1%) | 7622 (<1%) | 0.068 | 201 (<1%) | 188 (<1%) | 0.002 |
| Substance use disorders | 17754 (5%) | 380971 (18%) | 0.432 | 17754 (5%) | 17606 (5%) | 0.002 |
| Medications, n (%) | | | | | | |
| Medications (any) | 199678 (54%) | 1354075 (65%) | 0.229 | 199678 (54%) | 199156 (53%) | 0.003 |
| Alcohol deterrents | 513 (<1%) | 4977 (<1%) | 0.023 | 513 (<1%) | 324 (<1%) | 0.015 |
| Gabapentin | 9347 (3%) | 206179 (10%) | 0.309 | 9347 (3%) | 9179 (2%) | 0.003 |
| Naloxone | 105 (<1%) | 8814 (<1%) | 0.083 | 105 (<1%) | 199 (<1%) | 0.012 |
| Naltrexone* | 436 (<1%) | 4002 (<1%) | 0.019 | 436 (<1%) | 268 (<1%) | 0.015 |
| Opioid analgesics | 48633 (13%) | 619773 (30%) | 0.413 | 48633 (13%) | 49372 (13%) | 0.006 |
| Sedatives/hypnotics | 30780 (8%) | 316465 (15%) | 0.215 | 30780 (8%) | 30435 (8%) | 0.003 |
| Skeletal muscle relaxants | 21527 (6%) | 330189 (16%) | 0.327 | 21527 (6%) | 21466 (6%) | 0.001 |
| Procedures, n (%) | | | | | | |
| Anesthesia | 28046 (8%) | 251928 (12%) | 0.153 | 28046 (8%) | 27595 (7%) | 0.005 |
| Surgery | 175569 (47%) | 1250514 (60%) | 0.256 | 175569 (47%) | 175628 (47%) | <0.0001 |

Abbreviations: chiropractic spinal manipulation (CSM), standard deviation (SD), standardized mean deviation (SMD)

*Variable displayed for descriptive purposes–not matched

propensity matching, 14.05% (95% CI: 13.94–14.17%) of the CSM cohort received an oral opioid prescription, compared to 20.54% (95% CI: 20.41–20.67%) of the usual medical care cohort, translating to a RR of 0.68 (95% CI: 0.68–0.69; P < .00001).

**Negative control.** The likelihood of receiving radiology services during follow-up was similar between cohorts according to our pre-specified threshold (CSM: 56%, usual medical care: 66%), yielding a RR of 0.85 (95% CI: 0.85–0.85; $P$ < .00001). This outcome serves as another marker of success of propensity matching and is intentionally not directly related to our primary outcome.

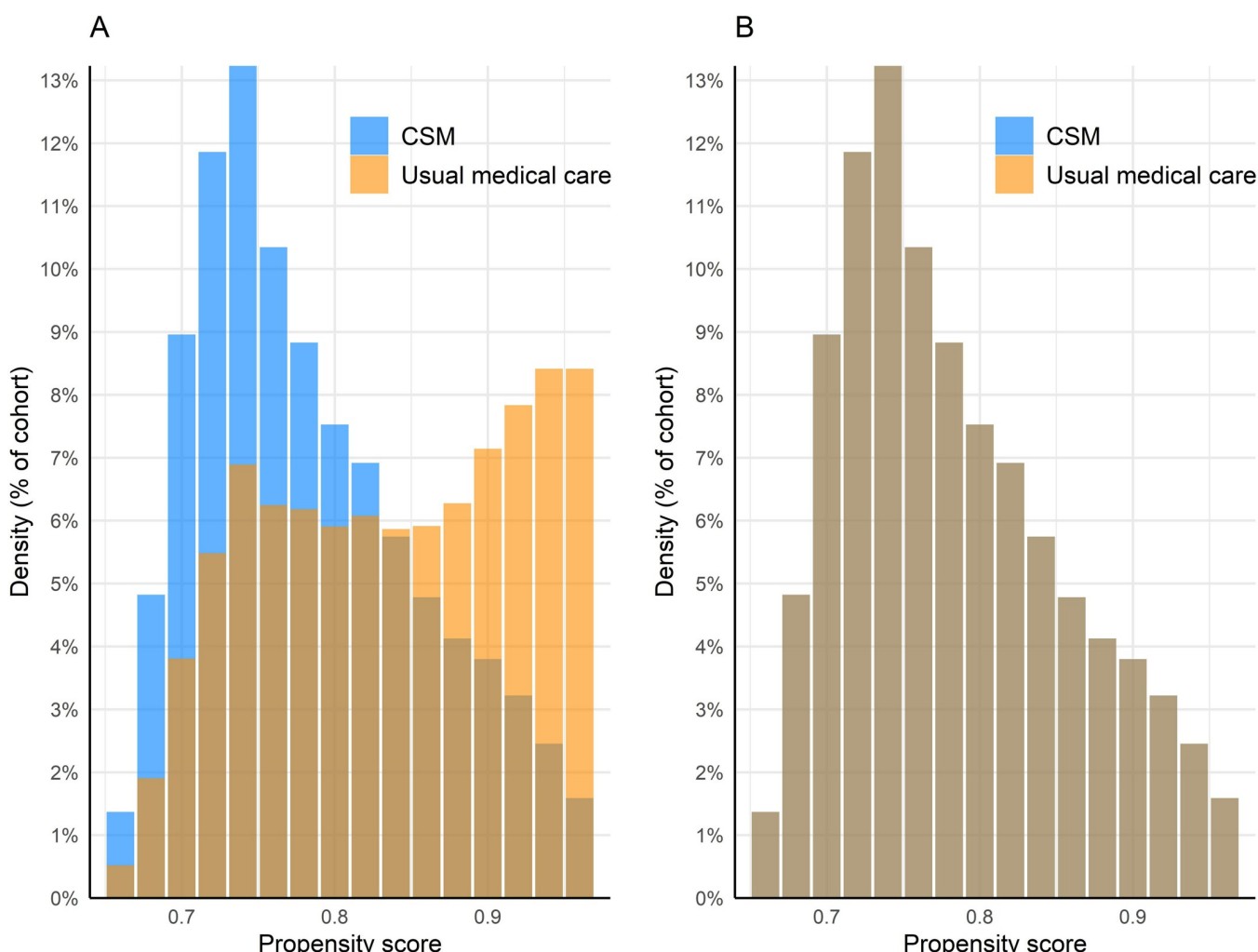

**Fig 1. Propensity score density plot.** Propensity scores are a maximum of 1.0 and are shown on the X-axis while the proportion of the cohort(s) is shown on the Y-axis. Density distributions of these scores are shown both preceding (A) and following (B) propensity score matching. Blue shading represents the chiropractic spinal manipulation (CSM) cohort, while orange represents the usual medical care cohort. The overlapping regions of blue and orange represent both cohorts. As a result of matching, propensity score densities appear superimposed, indicating successful balance of confounding variables.

**CSM visits.** Some crossover (also called contamination) from the usual medical care cohort into the CSM cohort during follow-up was evident and expected given our real-world approach that did not restrict patients to a defined protocol after the index date. After

**Table 2. Key outcomes for risk of opioid-related adverse events.**

|  | Before matching | | After matching | |
|---|---|---|---|---|
|  | CSM | Usual medical care | CSM | Usual medical care |
| Number of patients | 372471 | 2090255 | 372471 | 372471 |
| ORADE, n | 324 | 10726 | 324 | 1132 |
| ORADE, % (95% CI) | 0.09 (0.08–0.10) | 0.50 (0.50–0.52) | 0.09 (0.08–0.10) | 0.30 (0.29–0.32) |
| RR (95% CI) | 0.17 (0.15–0.19; P < .00001) | (reference) | 0.29 (0.25–0.32; P < .00001)* | (reference) |

Abbreviations: chiropractic spinal manipulation (CSM), opioid-related adverse event (ORADE), risk ratio (RR), 95% confidence intervals (95% CI)

*primary outcome

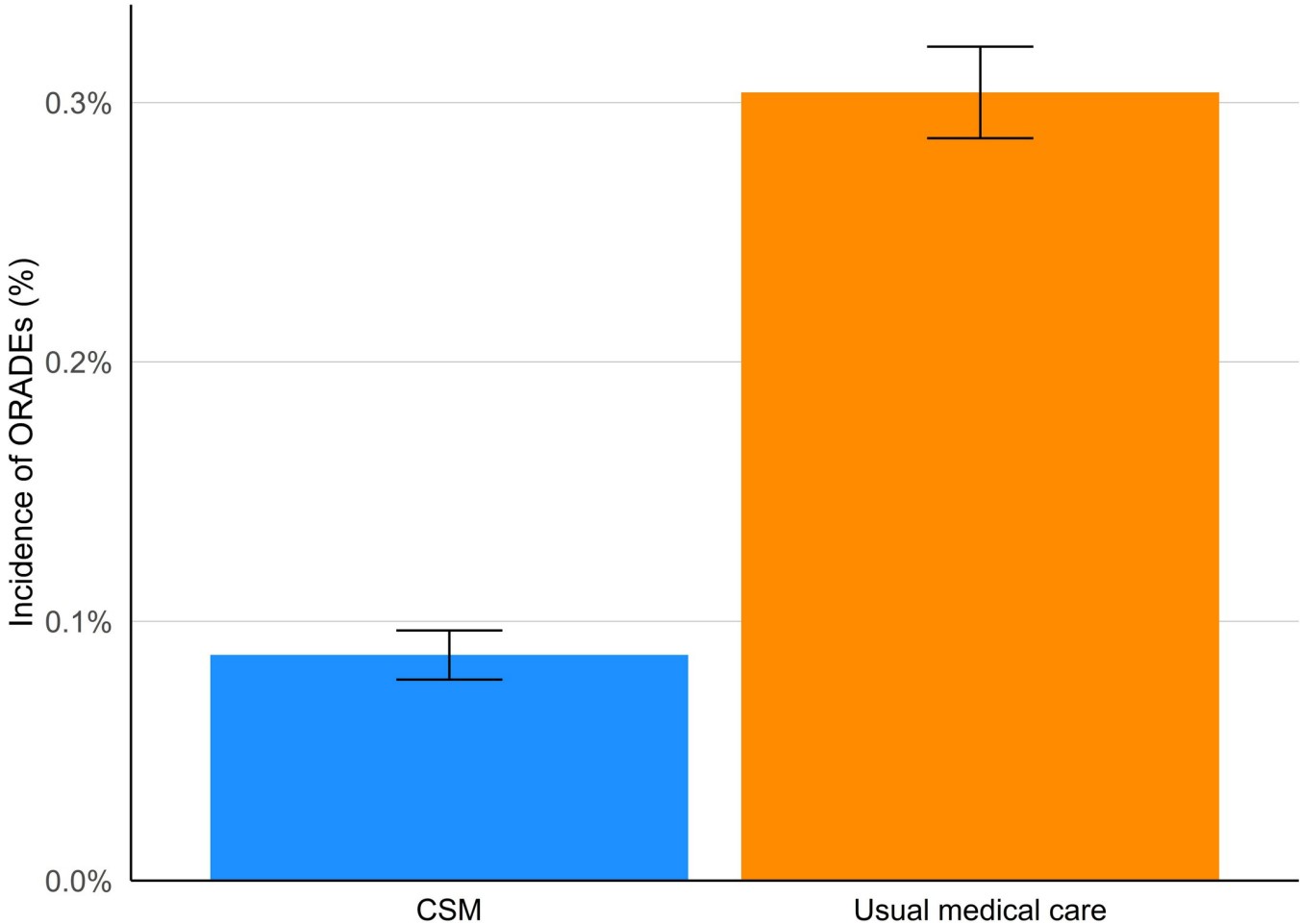

**Fig 2. Incidence of opioid-related adverse events (ORADEs) after propensity matching.** Incidence in the chiropractic spinal manipulation (CSM) cohort (blue) is lower than that of the usual medical care cohort (orange). The 95% confidence intervals do not overlap, suggesting a meaningful difference between cohorts.

matching, 86% of the CSM cohort had an additional CSM visit while 9% of the usual medical care cohort received CSM over the one-year follow-up window. Among those receiving CSM during follow-up, the mean number of CSM visits per cohort was similar [SD] (CSM: 8.9 [9.8]; usual medical care: 8.9 [9.9]). Compared to those receiving usual care, CSM patients had a significantly greater likelihood of receiving CSM during follow-up [95% CI] (RR = 9.12 [9.03, 9.21]; P<0.0001). Given that crossover would typically attenuate the observed effect estimate [41], our findings should not be explained by the small proportion of usual care cohort who received CSM during follow-up.

## Discussion

The present study supports our hypothesis that among adults with a new diagnosis of sciatica, initially receiving CSM is associated with a reduced risk of an ORADE over a one-year follow-up compared to matched controls not initially receiving CSM. To our knowledge, this study is the first to specifically examine this outcome. Analysis of cumulative incidence suggests that a detectable difference in risk of ORADEs begins immediately after the index date and persists for at least one year after follow-up. Our secondary outcome suggests that a reduced risk of an

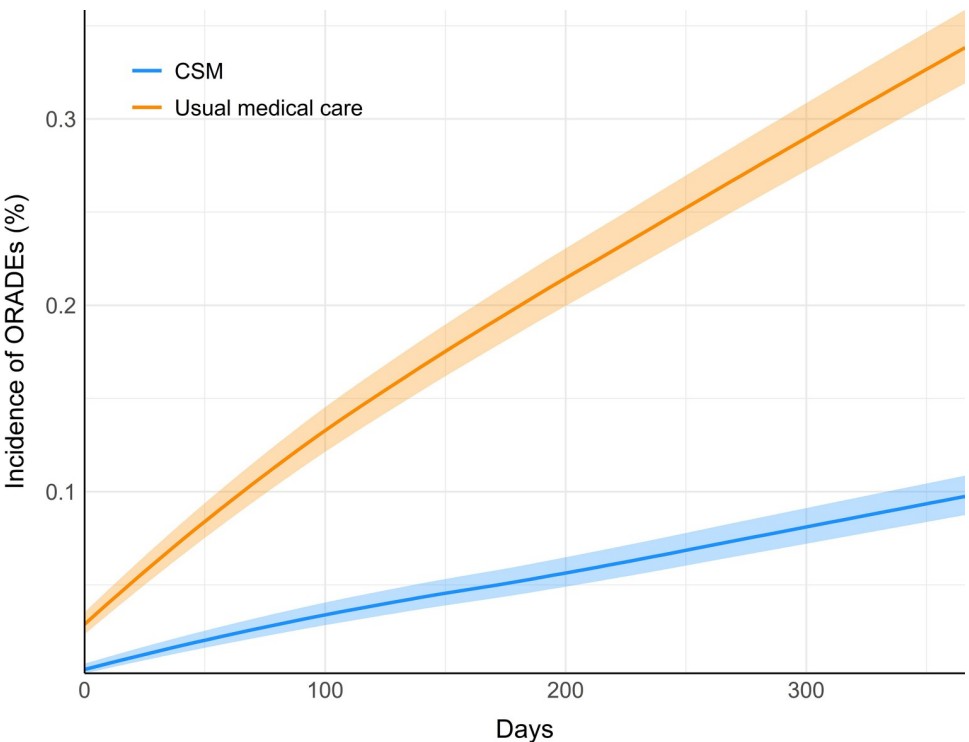

**Fig 3. Cumulative incidence of the occurrence of an opioid-related adverse event per patient in each cohort.** The incidence curve for the cohort receiving chiropractic spinal manipulation (CSM) is shown in orange, while that of the cohort receiving usual medical care is shown in blue, over the 365-day (one-year) follow-up window. Semi-transparent bands indicate 95% confidence intervals for each of the incidence curves in their respective colors.

ORADE in the CSM cohort may be, at least in part, attributed to a reduction in probability of oral opioid prescription.

The usual medical care cohort pre-matching incidence of ORADEs is the largest-powered and most realistic value to compare with previous estimates of this outcome in the general population. Our finding of an ORADE affecting 0.50% of patients in the usual medical care cohort can be translated to five ORADEs per 1000 patients. A recent meta-analysis which included over six million participants treated with opioids for chronic noncancer pain estimated a crude mortality of 1.1 per 1000 person-years (95% CIs: 0.4–3.4). Accordingly, our estimate falls within the range of plausible values of this meta-analysis [42]. However, a direct comparison between our estimate and this value is precluded by differences in the study population (chronic pain versus newly diagnosed sciatica) and outcome (mortality versus ORADEs, which are not all necessarily fatal).

Our novel finding of reduced risk of an ORADE among CSM recipients builds upon the prior literature on this topic. A retrospective observational study found that older adults aged 65–84 years (n = 28,160) who began opioid analgesic therapy for chronic low back pain had a significantly greater adjusted rate of compared to those who sought CSM (rate ratio of 42.85, 95% CI: 34.16–53.76, $P < .0001$) [22]. In addition to having a slightly different population (chronic symptoms, older individuals), this prior study focused on a variety of adverse drug events, rather than ORADEs only as in our study. A similar cohort study found that the risk of any adverse drug event was lower among CSM recipients versus nonrecipients among adults with low back pain (n = 19,153; odds ratio of 0.49; $P = .0002$) [21]. While the risk estimates from these previous studies are not directly comparable to our findings, a consistent theme has

emerged whereby patients initially receiving CSM for low back pain are less likely to have an adverse drug event.

Our findings have implications for both patients and clinicians. According to the CDC, clinicians prescribing opioids should consider the evidence, balance of desirable and undesirable effects, patient values and preferences, and resource allocation [7]. Accordingly, some patients are eager to avoid opioid prescriptions and ORADEs and thus may be advised regarding CSM as a viable care pathway for their symptoms. As CSM is already recommended by several clinical practice guidelines for sciatica to be used in conjunction with other therapies (e.g., exercise) [14–18], clinicians may consider CSM in appropriate clinical contexts (e.g., patient preference for CSM; lack of contraindications such as spinal infection, structural instability, or cauda equina syndrome [18, 43]). The present findings may also inform efforts by stakeholders when updating clinical practice guidelines for spinal disorders to consider therapies that are opioid-sparing [44].

Additional research should be performed to build on our findings. It remains plausible that any reduction in ORADEs identified in the present study may be attributable to an interaction with clinicians who offer non-pharmacological therapies (i.e., a chiropractor), rather than the CSM intervention itself [45]. Accordingly, additional study designs are warranted comparing a range of clinician types, such as acupuncturists, physical therapists, psychologists, primary care physicians, and medical specialists, which could uncover any potential broader influence of nonpharmacologic care. Additionally, the present study hypothesis could be tested in diverse subpopulations of low back pain, focusing on individuals at greater baseline risk of ORADEs, and/or using alternate study designs (e.g., case-control). Given the rarity of the outcome and large sample size required, a randomized controlled trial may be challenging to conduct.

## Strengths and limitations

Strengths of this study include an interdisciplinary author team, controlling for prior opioid prescriptions and other relevant ORADE risk factors, examination of adverse events specific to opioids, large sample size of 744,942 total patients, detailed selection criteria, and relatively long follow-up window.

Our observational study design presents limitations precluding our ability to infer causality. An inability to validate our query against a gold standard chart review introduces uncertainty in the accuracy of the data. Selection bias may be present with respect to the duration and potency of opioids prescribed (e.g., morphine equivalent daily dose) at baseline [46]. There may be residual confounding related to items unavailable in the dataset including: concurrent undocumented illicit substance use, number of unique opioid prescribers [47], severity of sciatic pain or functional impairment, unreported naloxone administration, and detailed socio-economic factors [7], or selection bias related to the availability of CSM [48]. Race and ethnicity are also poorly represented in the TriNetX Diamond Network. We were unable to examine whether ORADEs were fatal or nonfatal given the dataset constraints. Our outcome may have yielded false positives related to misdiagnosis of ORADEs, as well as false negatives due to unreported ORADEs occurring outside of a healthcare setting. As of September 2024, the TriNetX Diamond Network is no longer available and thus researchers who wish to replicate this study would need to use a different large claims-based data resource, such as Pearldiver or IBM Marketscan [49]. While replication using health records-based data is also possible, these may be limited by comparatively smaller sample sizes. Our findings may not be generalizable to conditions aside from sciatica and countries outside of the US which may have differences in prescribing patterns, access to naloxone products, and clinical triage approaches to ORADEs.

## Conclusions

We found that adults with a new diagnosis of sciatica who initially received CSM had a significantly lower risk of ORADEs over 1-year follow-up compared to matched controls initially receiving usual medical care. This finding was likely explained, in part, by a reduction in oral opioid prescription during follow-up. Our study builds on previous evidence to suggest that among those with spinal pain, upstream exposure to CSM may influence downstream use of opioids. The present findings suggest that CSM has value in terms of potential mitigation of ORADEs among those with sciatica and reinforces recommendations of previous guidelines to consider the use of CSM alongside other therapies for this condition.

## Supporting information

**S1 Table. Inclusion codes for both cohorts.**
(DOCX)

**S2 Table. Exclusion codes for both cohorts.**
(DOCX)

**S3 Table. Variables controlled for in propensity score matching.**
(DOCX)

**S4 Table. Opioid-related adverse drug events.**
(DOCX)

## Acknowledgments

The views expressed are those of the authors and do not necessarily reflect the official policy or position of the US Department of Veterans Affairs or the US Government.

## Author Contributions

**Conceptualization:** Robert J. Trager, Zachary A. Cupler, Roshini Srinivasan, Jaime A. Perez.

**Data curation:** Robert J. Trager, Elleson G. Harper, Jaime A. Perez.

**Formal analysis:** Robert J. Trager, Jaime A. Perez.

**Investigation:** Robert J. Trager, Zachary A. Cupler, Roshini Srinivasan, Jaime A. Perez.

**Methodology:** Robert J. Trager, Zachary A. Cupler, Roshini Srinivasan, Jaime A. Perez.

**Project administration:** Robert J. Trager, Jaime A. Perez.

**Resources:** Robert J. Trager, Elleson G. Harper, Jaime A. Perez.

**Software:** Robert J. Trager, Elleson G. Harper, Jaime A. Perez.

**Supervision:** Robert J. Trager, Jaime A. Perez.

**Visualization:** Robert J. Trager, Jaime A. Perez.

**Writing – original draft:** Robert J. Trager.

**Writing – review & editing:** Robert J. Trager, Zachary A. Cupler, Roshini Srinivasan, Elleson G. Harper, Jaime A. Perez.

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
