## [Decision Letter · Decision Letter 0]

9 Dec 2024

PONE-D-24-17756Association between chiropractic spinal manipulation for sciatica and opioid-related adverse events: a retrospective cohort studyPLOS ONE

Dear Dr. Trager,

Thank you for submitting your manuscript to PLOS ONE. After careful consideration, we feel that it has merit but does not fully meet PLOS ONE’s publication criteria as it currently stands. Therefore, we invite you to submit a revised version of the manuscript that addresses the points raised during the review process.

Thank you for your patience. After careful work to identify independent reviewers and carefully review your article, I invite you to adjust the article and resubmit it.

We look forward to receiving your revised manuscript.

Kind regards,

André Pontes-Silva

Academic Editor

PLOS ONE

Comments from Senior Staff Editor: We note that one or more reviewers has recommended that you cite specific previously published works. As always, we recommend that you please review and evaluate the requested works to determine whether they are relevant and should be cited. It is not a requirement to cite these works. 

Journal Requirements:

https://doi.org/10.1371/journal.pone.0299159

https://doi.org/10.1136/bmjopen-2023-078105

In your revision ensure you cite all your sources (including your own works), and quote or rephrase any duplicated text outside the methods section. Further consideration is dependent on these concerns being addressed.

3. Thank you for stating the following in the Competing Interests section: [I have read the journal's policy and the authors of this manuscript have the following competing interests: Robert J. Trager acknowledges that he has received royalties as the author of two texts on the topic of sciatica. The other authors have declared no competing interests].

Reviewers' comments:

Reviewer's Responses to Questions

**Comments to the Author**

1. Is the manuscript technically sound, and do the data support the conclusions?

Reviewer #1: Yes

Reviewer #2: Yes

Reviewer #3: Yes

2. Has the statistical analysis been performed appropriately and rigorously? 

Reviewer #1: I Don't Know

Reviewer #2: Yes

Reviewer #3: Yes

3. Have the authors made all data underlying the findings in their manuscript fully available?

Reviewer #1: Yes

Reviewer #2: Yes

Reviewer #3: Yes

4. Is the manuscript presented in an intelligible fashion and written in standard English?

Reviewer #1: Yes

Reviewer #2: Yes

Reviewer #3: Yes

5. Review Comments to the Author

Reviewer #1: In the Abstract, a description of the approach to comparative outcomes analysis is missing.

Page 3: RE: "...estimate also includes deaths from illicit use"

Many if not most of the deaths may be due to use of illicit fentanyl.

Page 4: RE: "We included data starting from 2009, 15 years prior to the query date (March 11, 2024), with the inclusion window ending one year prior to the query date to allow for ascertainment of the outcome."

This is a vague description. Define ”inclusion window”. If you mean you included patients with encounters for sciatica from 3/11/09 through 3/11/23, then say so.

Page 5. RE: "...require patients to have a pre- and post-index medical evaluation"

Define "pre- and post-index". The term "index" was not used prior to this instance.

Page 6: RE: "We included adults at least age 18 years, at the first occurrence of any diagnosis code of with sciatica or lumbosacral radiculopathy"

How did you identify "first occurrence"? Did all patients have a new episode of sciatica? Did the treatment on that date represent initial management?

Page 6: RE: "We excluded patients from the usual medical care cohort who received CSM on the index date of cohort eligibility."

and Page 7:RE: "Patients were divided into two cohorts dependent on the treatment received on the index date of sciatica diagnosis: (1) CSM; those receiving any CPT code for this procedure (98940, 98941, 98942); and

(2) usual medical care; those having an outpatient office visit (CPT: 1013625) and not receiving CSM on that date."

It appears that the CSM may have received medical care, and the medical care cohort may have received CSM, but not on the date of first occurrence.

What if patients in the medical care cohort received CSM the day before? A look-back period with no CSM (typically 30-90 days) (and a similar washout period for the CSM cohort) would establish "clean" mutually exclusive cohorts. Was that the intention? If not, OK, but then be careful about how you describe the cohorts, and how you express the results and conclusions. This point and the one immediately above about "first occurrence" are critical to the definition and description of cohorts.

Page 7: RE: "We propensity matched patients to reduce bias [23], balancing confounders present within a year preceding and including the date of inclusion associated with risk of ORADEs, including previous prescription medications and opioids (S3 Table)."

What about matching on other variables - health status, patient characteristics?

Page 14: RE: "The present study supports our hypothesis that adults receiving CSM for sciatica have a reduced risk of an ORADE over a one-year follow-up compared to matched controls not receiving CSM."

This conclusion appears to be unsupported, because as defined, the matched controls (medical care cohort) may have received CSM before or after the "first occurrence" date.

Page 17: RE: "We found that adults with sciatica who received CSM had a significantly lower risk of ORADEs over 1- year follow-up compared to matched controls receiving usual medical care."

This conclusion appears to be unsupported, because as defined, the matched controls (medical care cohort) may have received CSM before or after the "first occurrence" date.

Reviewer #2: The manuscript presents a technically sound study with robust statistical analyses that support the conclusions. The use of a large sample size (744,942 patients after matching) provides ample statistical power to detect meaningful differences between cohorts. The authors employ propensity score matching to address confounding, achieving balance in baseline characteristics as demonstrated by standardized mean differences below 0.1. This rigorous matching minimizes bias and enhances the validity of the comparisons.

Key outcomes, such as the reduced risk of opioid-related adverse events (ORADEs) in the chiropractic spinal manipulation (CSM) cohort (RR = 0.29, 95% CI: 0.25–0.32; P < 0.00001), are well-supported by clear and precise statistical analyses. Sensitivity analyses, including cumulative incidence plots, reinforce these findings, demonstrating consistent differences between cohorts throughout the follow-up period. The secondary outcome, which highlights a lower incidence of oral opioid prescriptions in the CSM cohort, provides a plausible mechanism for the observed reduction in ORADEs.

The authors also strengthen their approach by including a negative control analysis, which confirms the adequacy of the propensity matching. While the observational nature of the study precludes causal inference, the methods and results are robust, well-presented, and align with prior research. Overall, the manuscript demonstrates a high level of statistical rigor and provides valuable evidence supporting the integration of CSM into the management of sciatica.

Reviewer #3: I am not a statistician or methodologist however the methods and statistics make sense to me.

My review of this paper is presented from the perspective of a clinician researcher. Thus I found it to be an excellent piece with findings directly applicable to clinical practice. I have no biting criticism to make and find overall little with which to take issue or make suggestions.

I would question whether 'efficacy' on p3 should read as 'clinical effectiveness' as the reference provided states. Lewis RA, Williams NH, Sutton AJ, Burton K, Din NU, Matar HE, et al. Comparative clinical

effectiveness of management strategies for sciatica: Systematic review and network meta-analyses.

Spine J. 2015;15:1461–77.

As an Australian I note corresponding calls to update clinical practice guidelines here, for example:

Amorin-Woods, L. G. and B. L. Woods (2023). "It is Time to Update Australian Clinical Care Standards and Practice Recommendations for Management of Spinal pain: A Commentary." Chiropractic Journal of Australia 50(1): 83-97.

I commend the authors on this work.

6. PLOS authors have the option to publish the peer review history of their article (what does this mean?). If published, this will include your full peer review and any attached files.

Reviewer #1: No

Reviewer #2: **Yes: **Benjamin Eovaldi, DO, MPH

Reviewer #3: No

---

## [Author Response · Author response to Decision Letter 0]

17 Dec 2024

Response to Reviewers - please note that we uploaded this as a separate attachment where the formatting may be easier to read.

Overview

We are deeply thankful for the reviewers thoughtful insights which have greatly strengthened our manuscript. We made edits to better describe the statistical comparison in the abstract, clarify the study data range, use of a first diagnosis of sciatica, and added detail regarding the crossover of usual care patients into the chiropractic spinal manipulation (CSM) cohort and described how this would not explain our findings. We improved our concluding statements to emphasize initial care received given the flexibility of our real-world design, and reduced text redundancy with prior work, among other minor changes.

Editorial comments

https://doi.org/10.1371/journal.pone.0299159

https://doi.org/10.1136/bmjopen-2023-078105

In your revision ensure you cite all your sources (including your own works), and quote or rephrase any duplicated text outside the methods section. Further consideration is dependent on these concerns being addressed.

a. Response: We are grateful for you bringing this to our attention. To gather more insight, we used iThenticate to identify segments of text that were similar. It appears much of the similar text appears in our Methods because the overlapping studies relied on similar statistical techniques, and used the same dataset and software, and were conducted by the same lead author as the present study. Some other sections were structured or worded similarly. We made several changes to alter the wording of the present manuscript to reduce duplicate text.

b. A few items could not be changed. There is a specific Acknowledgement that we made in this manuscript as well as the BMJ Open manuscript that is specific to the US Department of Veterans Affairs, as well as a funding statement and competing interest statement that should remain similar as some of the same authors were involved.

c. Overlap between the current manuscript and prior PLOS ONE and BMJ Open manuscript (overlap with both)

i. Methods statements that were reworded (see tracked changes in manuscript):

1. “Study reporting adheres to the Strengthening the Reporting of Observational Studies in Epidemiology (STROBE) guideline” now states “The findings are reported according to the Strengthening the Reporting of Observational Studies in Epidemiology (STROBE)”

2. “We calculated a total required sample size of” now states “We estimated a required total sample size”

ii. Results sections that were reworded due to similar phrasing of text or paragraph structure (see tracked changes):

1. Descriptive data – we reworded this paragraph as it was structured similarly with some of the same terminology as previous studies.

2. Figure 1 – we reworded the caption as the phrasing overlapped with previous studies.

d. Current manuscript versus prior PLOS ONE manuscript: 

i. Methods sections that were reworded due to similar phrasing of text (see tracked changes in manuscript for edits):

1. “This study implemented a retrospective cohort design with active comparator features to minimize bias” now states “The present retrospective cohort study incorporated active comparator features to minimize bias”

2. HIPAA compliance disclaimer – The section regarding privacy and de-identification was both shortened and reworded. 

3. Statistical methods – We reworded several aspects of this paragraph to minimize overlap with prior studies.

ii. Results sections that were reworded due to similar phrasing of text:

1. Figure 3 – We reworded the caption.

3. Thank you for stating the following in the Competing Interests section: [I have read the journal's policy and the authors of this manuscript have the following competing interests: Robert J. Trager acknowledges that he has received royalties as the author of two texts on the topic of sciatica. The other authors have declared no competing interests].

a. Thank you. We have added this statement as advised: “This does not alter our adherence to PLOS ONE policies on sharing data and materials”

a. Response: Thank you. We have reviewed the reference list using Zotero which has a function to flag retracted articles and we found none that were retracted. We also added a reference as advised by Reviewer #3.

Comments to the Author

1. Is the manuscript technically sound, and do the data support the conclusions?

Reviewer #1: Yes

Reviewer #2: Yes

Reviewer #3: Yes

2. Has the statistical analysis been performed appropriately and rigorously?

Reviewer #1: I Don't Know

Reviewer #2: Yes

Reviewer #3: Yes

3. Have the authors made all data underlying the findings in their manuscript fully available?

Reviewer #1: Yes

Reviewer #2: Yes

Reviewer #3: Yes

4. Is the manuscript presented in an intelligible fashion and written in standard English?

Reviewer #1: Yes

Reviewer #2: Yes

Reviewer #3: Yes

Review Comments to the Author

Reviewer 1

1. In the Abstract, a description of the approach to comparative outcomes analysis is missing.

a. Response: Thank you, we agree and have added the following to the abstract: Comparative outcomes were analyzed by calculating risk ratios (RRs) and 95% confidence intervals (CIs) for the incidence of ORADEs and oral opioid prescription between cohorts.

2. Page 3: RE: "...estimate also includes deaths from illicit use"

Many if not most of the deaths may be due to use of illicit fentanyl.

a. Response: Thank you. We have modified the current statement to be more straightforward in noting that fentanyl plays a major role in influencing these statistics: although this estimate also includes deaths from non-prescription opioid use, particularly illicit fentanyl [1], which influences mortality statistics. As a result, this value does not represent deaths strictly from prescribed opioids.

3. Page 4: RE: "We included data starting from 2009, 15 years prior to the query date (March 11, 2024), with the inclusion window ending one year prior to the query date to allow for ascertainment of the outcome."

This is a vague description. Define ”inclusion window”. If you mean you included patients with encounters for sciatica from 3/11/09 through 3/11/23, then say so.

a. Response: Thank you for this idea. We have improved the statement as follows and have avoided the term “inclusion window” which was vague and confusing: “Patients meeting the selection criteria from March 11, 2009, through March 11, 2023, were included, ensuring a one-year follow-up period for outcome ascertainment prior to the query date of March 11, 2024.”

4. Page 5. RE: "...require patients to have a pre- and post-index medical evaluation"

Define "pre- and post-index". The term "index" was not used prior to this instance.

a. Response: Thank you. We have now edited this as follows, avoiding the phrase “index date” which is not defined until later: require patients to have a medical evaluation before and after the date of inclusion in the study.

5. Page 6: RE: "We included adults at least age 18 years, at the first occurrence of any diagnosis code of with sciatica or lumbosacral radiculopathy"

How did you identify "first occurrence"? Did all patients have a new episode of sciatica? Did the treatment on that date represent initial management?

a. Response: Thank you for the opportunity to expand on this. While we could have defined our cohorts as having a new episode of sciatica, we instead opted for a stricter definition, identifying the first-ever recorded diagnosis of sciatica. While this may not be perfect, as patients can avoid seeing doctor for several weeks, we do note the general limitations of our query definition in the Discussion. In general, our strict approach minimizes the likelihood of including patients with chronic symptoms prior to inclusion and standardizes the cohorts for better comparability [2].

b. Added text: We used the first recorded diagnosis code of sciatica as the index date rather than relying on a specific washout period, with the aim of minimizing imbalance between cohorts’ durations of sciatica.

6. Page 6: RE: "We excluded patients from the usual medical care cohort who received CSM on the index date of cohort eligibility."

and Page 7:RE: "Patients were divided into two cohorts dependent on the treatment received on the index date of sciatica diagnosis: (1) CSM; those receiving any CPT code for this procedure (98940, 98941, 98942); and

(2) usual medical care; those having an outpatient office visit (CPT: 1013625) and not receiving CSM on that date."

It appears that the CSM may have received medical care, and the medical care cohort may have received CSM, but not on the date of first occurrence.

What if patients in the medical care cohort received CSM the day before? A look-back period with no CSM (typically 30-90 days) (and a similar washout period for the CSM cohort) would establish "clean" mutually exclusive cohorts. Was that the intention? If not, OK, but then be careful about how you describe the cohorts, and how you express the results and conclusions. This point and the one immediately above about "first occurrence" are critical to the definition and description of cohorts.

a. Response: Thank you for this insightful comment. You are correct that there is potential for some crossover (also called contamination) between cohorts outside of the index date when patients were included. While this may represent a limitation in randomized controlled studies, we believe it is acceptable given the real-world observational nature of our study design by allowing for greater generalizability. Despite the crossover, which typically attenuates the magnitude of associations (i.e., regression to the null) [3], we still observed a significant difference between cohorts. 

b. The issue of crossover is best studied in relation to clinical trials, wherein a greater proportion of crossover necessitates a larger sample size. While up to 30% crossover may be acceptable [4], this varies contextually. Fortunately, in our study the large sample size (total n=744,942) appears to have mitigated the attenuating effects of crossover. In addition, the proportion of crossover was relatively small. Only 9% of the usual medical care cohort received CSM over the 1-year follow-up.

c. Before the index date: Considering our index date of including patients focused on new diagnoses of sciatica, patients in the usual medical care cohort who may have received chiropractic care prior to this likely sought it for conditions other than sciatica (e.g., neck pain or headache). Accordingly, we feel that receiving CSM for other conditions would not act as a substantial confounding factor.

d. If we had restricted the usual medical care cohort from receiving CSM after the index date, this could have led to biases related to poorer healthcare engagement or lower access to care. 

e. Also, please note that the Diamond network has been sunset as of September 2024 and no longer accessible. Therefore, we cannot explore the real-time data to determine the proportion of patients in the usual care cohort who received CSM prior to the index date of our present study. 

f. We have added a statement about the proportion of the usual medical care cohort receiving CSM in the Results / Secondary Outcomes as follows, as well as statements reflective of the overall CSM care across both cohorts as follows: Some crossover (also called contamination) from the usual medical care cohort into the CSM cohort during follow-up was evident and expected given our real-world approach that did not restrict patients to a defined protocol after the index date. After matching, 86% of the CSM cohort had an additional CSM visit while 9% of the usual medical care cohort received CSM over the one-year follow-up window. Among those receiving CSM during follow-up, the mean number of CSM visits per cohort was similar [SD] (CSM: 8.9 [9.8]; usual medical care: 8.9 [9.9]). Compared to those receiving usual care, CSM patients had a significantly greater likelihood of receiving CSM during follow-up [95% CI] (RR=9.12 [9.03, 9.21]; P<0.0001). Given that crossover would typically attenuate the observed effect estimate [3], our findings should not be explained by the small proportion of usual care cohort who received CSM during follow-up.

g. We have added a limitation as follows: As of September 2024, the TriNetX Diamond Network is no longer available and thus researchers who wish to replicate this study would need to use a different large claims-based data resource, such as Pearldiver or IBM Marketscan [5]. While replication using health records-based data is also possible, these may be limited by comparatively smaller sample sizes.

7. Page 7: RE: "We propensity matched patients to reduce bias [23], balancing confounders present within a year preceding and including the date of inclusion associated with risk of ORADEs, including previous prescription medications and opioids (S3 Table)."

What about matching on other variables - health status, patient characteristics?

a. Response: We appreciate this insightful question. Our p

---

## [Editor Report · Decision Letter 1]

3 Jan 2025

Association between chiropractic spinal manipulation for sciatica and opioid-related adverse events: a retrospective cohort study

PONE-D-24-17756R1

Dear Dr. Robert James Trager,

We’re pleased to inform you that your manuscript has been judged scientifically suitable for publication and will be formally accepted for publication once it meets all outstanding technical requirements.

Kind regards,

André Pontes-Silva

Academic Editor

PLOS ONE
---

## [Editor Report · Acceptance letter]

17 Jan 2025

PONE-D-24-17756R1 

PLOS ONE

Dear Dr. Trager, 

I'm pleased to inform you that your manuscript has been deemed suitable for publication in PLOS ONE. Congratulations! Your manuscript is now being handed over to our production team.

Kind regards, 

on behalf of

Professor André Pontes-Silva 

Academic Editor

PLOS ONE